# Increase in Anticancer Drug-Induced Toxicity by Fisetin in Lung Adenocarcinoma A549 Spheroid Cells Mediated by the Reduction of Claudin-2 Expression

**DOI:** 10.3390/ijms23147536

**Published:** 2022-07-07

**Authors:** Hiroaki Eguchi, Riho Kimura, Haruka Matsunaga, Toshiyuki Matsunaga, Yuta Yoshino, Satoshi Endo, Akira Ikari

**Affiliations:** 1Laboratory of Biochemistry, Department of Biopharmaceutical Sciences, Gifu Pharmaceutical University, Gifu 501-1196, Japan; 146008@gifu-pu.ac.jp (H.E.); 185039@gifu-pu.ac.jp (R.K.); 165077@gifu-pu.ac.jp (H.M.); yoshino-yu@gifu-pu.ac.jp (Y.Y.); sendo@gifu-pu.ac.jp (S.E.); 2Education Center of Green Pharmaceutical Sciences, Gifu Pharmaceutical University, Gifu 502-8585, Japan; matsunagat@gifu-pu.ac.jp

**Keywords:** lung adenocarcinoma, fisetin, anticancer drug toxicity, claudin-2

## Abstract

Claudin-2 (CLDN2), a component of tight junction, is involved in the reduction of anticancer drug-induced toxicity in spheroids of A549 cells derived from human lung adenocarcinoma. Fisetin, a dietary flavonoid, inhibits cancer cell growth, but its effect on chemosensitivity in spheroids is unknown. Here, we found that fisetin (20 μM) decreases the protein level of CLDN2 to 22.3%. Therefore, the expression mechanisms were investigated by real-time polymerase chain reaction and Western blotting. Spheroids were formed in round-bottom plates, and anticancer drug-induced toxicity was measured by ATP content. Fisetin decreased the phosphorylated-Akt level, and CLDN2 expression was decreased by a phosphatidylinositol 3-kinase (PI3K) inhibitor, suggesting the inhibition of PI3K/Akt signal is involved in the reduction of CLDN2 expression. Hypoxia level, one of the hallmarks of tumor microenvironment, was reduced by fisetin. Although fisetin did not change hypoxia inducible factor-1α level, it decreased the protein level of nuclear factor erythroid 2-related factor 2, a stress response factor, by 25.4% in the spheroids. The toxicity of doxorubicin (20 μM) was enhanced by fisetin from 62.8% to 40.9%, which was rescued by CLDN2 overexpression (51.7%). These results suggest that fisetin can enhance anticancer drug toxicity in A549 spheroids mediated by the reduction of CLDN2 expression.

## 1. Introduction

Lung cancer is the most common malignancy worldwide and the leading cause of cancer death. Novel anticancer drugs including targeted molecule agents and immune checkpoint modulators have improved patient survival [1,2]. However, the problems of acquisition of chemoresistance and recurrence of cancers remain unresolved. The reduction of chemosensitivity is caused by various mechanisms including the immune system suppression; activation of DNA repair system; and induction of drug efflux transporters, metabolic enzymes, and antioxidant enzymes [3]. Cancer cells form the microenvironment together with endothelial cells, fibroblasts, immune cells, and stem cells in the body [4]. The tumor microenvironment is characterized by regions of insufficient oxygen (hypoxia), nutrient deprivation, and acidic pH. Hypoxia induces reactive oxygen species (ROS) generation and increases nuclear factor erythroid 2-related factor 2 (Nrf2) activity [5]. These stress conditions can facilitate the acquisition of chemoresistance in cancer cells and survival of cancer stem cells.

Epithelial cells form semi-permeable paracellular barriers at the cell–cell contact area and control the permeability to mineral ions, low molecular compounds, and water [6]. The essential structure forming the barrier is a tight junction (TJ) [7]. The TJ is located at the most apical side of the lateral membranes and the core transmembrane proteins are claudins (CLDNs), which comprise a large family of 27 subtypes [8,9]. Among them, CLDN2 is mainly expressed in the leaky epithelia of the renal tubules and small intestine under physiological conditions and forms a paracellular cation channel permeating sodium ion and water [10,11]. On the other hand, CLDN1 is widely expressed in the various tissues and function as barrier to small molecules. The upregulation of CLDN2 expression has been reported in various cancer tissues such as liver [12], breast [13], endometrium [14], and stomach [15]. CLDN2 facilitates the tumorigenicity by enabling anchorage-independent growth in the colorectal cancer cells [16], which is tightly correlated with cancer progression. We previously reported that CLDN2 expression is upregulated in the malignant tissues and cell lines derived from human lung adenocarcinoma [17]. The downregulation of CLDN2 expression by siRNA inhibits proliferation and migration in monolayer culture model of human lung adenocarcinoma A549 cells [18,19]. Furthermore, hypoxic stress and chemoresistance against anticancer drugs in the spheroids are attenuated by the suppression of CLDN2 expression [20]. Therefore, CLDN2 may be a novel target for reversing chemoresistance in lung adenocarcinoma.

Many flavonoids have been reported to have antioxidant, anticarcinogenic, anti-inflammatory, and antiviral activities [21]. Flavonoids possess a common structure consisting of two aromatic rings connected to three carbon atoms. In general, it is considered that the number of hydroxy substituents is correlated with the antioxidant activity. In contrast, the correlation between the structure and other functions of flavonoids are not fully understood. So far, we reported that some flavonoids including quercetin, kaempferol, and kaempferide can reduce CLDN2 expression and enhance anticancer drug-induced toxicity in A549 cells [22,23,24]. The effect of kaempferide on CLDN2 expression is higher than that of kaempferol, although kaempferol has a potent antioxidant ability. Furthermore, *N*-acetyl-L-cysteine, a pharmacological antioxidant, does not have the ability to decrease CLDN2 expression [22]. Other regulatory mechanisms without an antioxidant effect may be involved in the flavonoid-induced reduction of CLDN2 expression.

In the present study, we found that fisetin, a flavonoid widely distributed in fruits and vegetables, can decrease CLDN2 expression in A549 cells. Therefore, the effects of fisetin on the expression regulation and function of CLDN2 were investigated by real-time polymerase chain reaction (PCR) and Western blotting analyses. The effects of fisetin on hypoxic stress and anticancer drug toxicity were investigated using the three-dimensional cultured spheroid cells.

## 2. Results

### 2.1. Decrease in Protein Level of CLDN2 in A549 Cells by Fisetin

A549 cells cultured on flat-bottom plates were incubated in the presence and absence of fisetin for 24 h. Cell viability was significantly decreased at the concentration over 10 μM, but the amount of change was less than 20% (Figure 1A). The protein level of CLDN2 was dose-dependently decreased by fisetin, whereas that of CLDN1 was not (Figure 1B,C). The protein level of CLDN2 was decreased to 22.3% by fisetin at 20 μM. These results indicate that the fisetin-induced decrease in CLDN2 expression may be independent of cytotoxicity. CLDN1 was localized at the cell–cell contact area concomitant with zonula occludens-1 (ZO-1), a scaffolding protein of TJs, under control conditions (Figure 1D). The distribution of CLDN1 was unchanged by fisetin. In contrast, CLDN2 was localized at the TJs under control conditions, but the fluorescence signal of CLDN2 disappeared from the TJs in the fisetin-treated cells. These results are identical to that of Western blotting (Figure 1B,C) and indicate that fisetin may decrease in CLDN2 expression without affecting CLDN1 expression in A549 cells.

### 2.2. Effect of Fisetin on Intracellular Signaling Factors

The expression of CLDN2 in A549 cells is upregulated by mitogen-activated protein kinase (MEK)/extracellular signal-regulated kinase 1/2 (ERK), PI3K/Akt, and Stat3 pathways [17,22,24]. Fisetin decreased the p-Akt (Thr308) and p-Akt (Ser473) levels without affecting total amounts of Akt (Figure 2A). In contrast, p-ERK1/2 and p-Stat3 levels were not significantly decreased by fisetin (Figure 2B). Phosphatidylinositol 3-kinase (PI3K) activates the phosphorylation of Akt mediated by phosphoinositide-dependent kinase 1 (PDK1). Neither p-PI3K nor p-PDK1 levels were changed by fisetin (Figure 2C).

### 2.3. Effects of Fisetin and PI3K Inhibitor on the Expression of CLDN2 mRNA

The effect of fisetin on mRNA expression and transcriptional activity of CLDN2 was investigated by real-time PCR and luciferase reporter assay, respectively. The mRNA level of CLDN2 was decreased by both fisetin and LY-294002, a PI3K inhibitor (Figure 3A). In contrast, that of CLDN1 was significantly changed by neither fisetin nor LY-294002. The reporter activity of CLDN2 was significantly decreased by both fisetin and LY-294002 (Figure 3B). These results indicate that fisetin may reduce the protein level of CLDN2 mediated by the inhibition of Akt-dependent transcriptional activity.

### 2.4. Effects of Fisetin on Property of Spheroids

So far, we reported that the anticancer drug toxicity of A549 spheroid cells is reduced by CLDN2 expression [20]. The expression of CLDN2 in A549 spheroid cells was significantly decreased by fisetin (Figure 4A), which is similar to that in 2D culture (Figure 1B,C). Neither spheroid size nor ATP content were changed by the treatment with fisetin (Figure 4B,C). In contrast, the fluorescence intensity of LOX-1, a hypoxia probe, was slightly decreased by fisetin. Hypoxia stress can upregulate the expression of stress response factors such as hypoxia inducible factor-1α (HIF-1α) and Nrf2 [5]. Fisetin (20 μM) decreased the protein level of Nrf2 by 25.4%, whereas it did not significantly change that of HIF-1α (Figure 4D). The mRNA levels of the ATP binding cassette (ABC) transporter B1 (ABCB1) and ABCC1 were significantly decreased by fisetin, whereas those of ABCC2 and ABCG2 were unchanged (Figure 4E).

### 2.5. Enhancement of Anticancer Drug-Induced Toxicity by Fisetin in Spheroids

After treating with doxorubicin (DXR), the fluorescence intensity of DXR in spheroids was dose-dependently increased (Figure 5A). The accumulation of DXR (20 μM) was enhanced by fisetin from 143.2% to 157.5%. In addition, the DXR-induced toxicity was dose-dependently increased, which was exaggerated by fisetin (Figure 5B). The DXR (20 μM)-induced toxicity was enhanced by fisetin from 37.4% to 41.6%. To clarify the mechanism of toxicity, the proportions of apoptotic and necrotic cell deaths were examined using a fluorescence microscope. The percentages of both apoptotic and necrotic cell deaths were increased by DXR compared to control from 3.8% to 8.2% (apoptosis) and from 5.3% to 11.7% (necrosis) (Figure 5C). Fisetin increased the percentages of both DXR-induced apoptotic (9.1%) and necrotic cell deaths (14.6%). Next, we investigated the effects of other anticancer drugs on viability of spheroid cells. The viabilities were dose-dependently decreased by cisplatin (CDDP), docetaxel (DOC), and gefitinib (GEF), which were significantly exaggerated by fisetin (Figure 6). These results are similar to those of DXR.

### 2.6. Attenuation of Fisetin-Induced Accumulation and Toxicity of DXR by CLDN2 Overexpression

As shown above, fisetin decreased CLDN2 expression (Figure 1B,C and Figure 4A) and increased the DXR-induced toxicity. To clarify the involvement of CLDN2 in the elevation of DXR-induced toxicity (Figure 5B), we investigated the effect of CLDN2 overexpression. The accumulation of DXR (20 μM) in the spheroid was significantly increased by fisetin from 111.0% to 122.6%, which was inhibited by CLDN2 overexpression (114.6%) (Figure 7). In addition, the DXR-induced reduction of cell viability was significantly enhanced by DXR (20 mM) from 62.8% to 40.9%, which was rescued by CLDN2 overexpression (51.7%). These results indicate that fisetin may enhance anticancer drug-induced toxicity mediated by the reduction of CLDN2 expression in A549 spheroid cells. The relationship between structure and function is summarized in Figure 8.

## 3. Discussion

CLDN2 is highly expressed in human lung adenocarcinoma A549, RERF-LC–MS (JCRB0081, JCRB Cell Bank), and PC-3 (JCRB0077, JCRB Cell Bank) cells and involved in the regulation of cell proliferation, migration, and anticancer drug toxicity [17,25]. Cell viability was partially suppressed by fisetin at the concentration of 10 and 20 μM (Figure 1A). Fisetin has been reported to suppress the growth and migration of A549 cells mediated by the inhibition of ERK1/2 pathway [26]. However, the inhibition of ERK1/2 was not observed under our experimental conditions (Figure 2). This discrepancy may be caused by the difference in treatment concentration. Fisetin interacts with multiple signaling pathways in various cancer cells leading to apoptosis and autophagic cell death [27]. Fisetin has a higher scavenging activity and defends the cells against ROS [28]. In contrast, fisetin can also increase the production of ROS and induce cytotoxic effects [29]. The fisetin-induced decrease in viability of A549 cells may be caused by excessive oxidative stress.

Some flavonoids including quercetin, kaempferol, and so on, can decrease CLDN2 expression in A549 cells [23,24]. However, the relationship between structure and function is not fully understood. In the present study, we found that fisetin, which is also called as 5-deoxyquercetin, can decrease CLDN2 expression (Figure 1B). The relationship between structure and function is summarized in Figure 8. CLDN2 expression has not been reported to be decreased by dihydrokaempferol, dihydrokaempferide, hesperetin, and pinocembrin. As compared to other flavonoids, the number and position of hydroxy and methoxy substituents in A, B, and C rings may not affect the ability to reduce CLDN2 expression. On the other hand, the 2,3-single bond in the C ring produces stereoisomers, which may have difference in biological activity. The 2,3-double bond in the C ring may be necessary for the reduction of CLDN2 expression by flavonoid derivatives.

The expression of CLDN2 in A549 cells is upregulated by several intracellular signaling pathways including MEK/ERK, PI3K/Akt, and Stat3 cascades [17,22,24]. The phosphorylation levels of Akt were significantly decreased by fisetin, whereas those of ERK1/2 and Stat3 were not (Figure 2), indicating that fisetin may suppress CLDN2 expression mediated by the inhibition of PI3K/Akt pathway. The inhibition effect of fisetin on PI3K/Akt signal has been reported in pancreatic [30], breast [31], and prostate cancer cells [32] as well as lung carcinoma A549 and H1792 cells [33]. The phosphorylation levels of PI3K and PDK1, upstream kinases of Akt, were not significantly inhibited by fisetin (Figure 2). The reporter activity of CLDN2 was inhibited by fisetin similar to LY-294002 (Figure 3). We suggest that fisetin decreases CLDN2 expression mediated by the inhibition of Akt-dependent transcriptional regulation.

In 3D culture systems, A549 cells grow as cellular aggregates called as spheroids which mimic in vivo organs. The sensitivity of spheroid cells against anticancer drugs is weaker than that of monolayer cells [20]. Hypoxia is one of the hallmarks of tumor microenvironment and a critical cause of chemoresistance. The hypoxia level was decreased by fisetin, but the protein level of HIF-1α was not in A549 spheroid cells (Figure 4). On the other hand, the protein level of Nrf2 was decreased by fisetin. Nrf2 is involved in the counteraction of oxidative modifications during hypoxia and enhances the expression of a number of stress-responsive genes that control survival, metabolic reprogramming, proliferation, and drug resistance [34,35]. ABCB1, ABCC1, and ABCG2 are highly expressed in the cisplatin-resistant A549 cells compared to parent cells [36]. Nrf2 promotes transcriptional activation of ABC transporters [34]. In addition, ABCC1 expression is decreased by transfection of Nrf2 shRNA in the cisplatin-resistant A549 cells [37]. Fisetin may enhance anticancer drug toxicity mediated by the inhibition of Nrf2-induced expression of ABCB1 and ABCC1 transporters (Figure 4E). Although ABCG2 is also a target gene of Nrf2, its expression was not decreased by fisetin. Other regulatory factors may be involved in the regulation of ABCG2 transporter.

The accumulation and sensitivity of DXR in spheroid cells were enhanced by the cotreatment with fisetin (Figure 5). DXR is a cytotoxic drug the metabolism that yields high amounts of ROS such as free oxygen radicals [38]. The generation of ROS by DXR induces apoptotic and necrotic cell deaths [39]. CLDN2 forms paracellular barrier to low molecular compounds including DXR [20]. The sensitivity of A549 spheroid cells against anticancer drugs including CDDP, DOC, and GEF was enhanced by fisetin. Furthermore, the effect of fisetin was cancelled by the overexpression of CLDN2 (Figure 7). In the electron microscope observation and immunofluorescence analysis, CLDN2 is abundantly expressed in the outer layer of spheroids and forms TJ [40]. Fisetin may also enhance the anticancer drug toxicity mediated by the increase in penetration of anticancer drugs in the spheroids.

## 4. Materials and Methods

### 4.1. Materials

Rabbit anti-p-Akt (Ser473), anti-p-Akt (Thr308), anti-Akt, anti-PDK1, anti-p-PDK1, anti-PI3K p85, anti-p-PI3K p85, and anti-ERK1/2 antibodies were purchased from Cell Signaling Technology (Beverly, MA, USA). Mouse anti-CLDN2, rabbit anti-CLDN2, mouse anti-ZO-1, and rabbit anti-ZO-1 antibodies were from Thermo Fisher Scientific (Rockford, IL, USA). Mouse anti-p-Stat3 (Y705) and anti-Stat3 antibodies were from BD Biosciences (Franklin Lakes, NJ, USA). Goat anti-β-actin and rabbit anti-p-ERK1/2 antibodies were from Santa Cruz Biotechnology (Santa Cruz, CA, USA). Fisetin, GEF, and LY-294002 were obtained from Cayman Chemical (Ann Arbor, MI, USA) and dissolved in dimethyl sulfoxide (DMSO). Control cells were treated with DMSO as a vehicle. The concentration of DMSO in the control and drug-treated cells was 0.1%. CDDP and DXR were from Fujifilm Wako Pure Chemical Industries (Osaka, Japan). DOC and hypoxia probe solution (LOX-1) were from Tokyo Chemical Industry (Tokyo, Japan) and Medical and Biological Laboratory (Tokyo, Japan), respectively. All other reagents were of the highest purity available.

### 4.2. Cell Culture and Transfection

A549 cells derived from human lung adenocarcinoma (RIKEN BRC through the National Bio-Resource Project of the MEXT, Ibaraki, Japan) were cultured in Dulbecco’s modified Eagle’s medium (DMEM, Sigma-Aldrich, St. Louis, MO, USA) supplemented with 5% fetal bovine serum (FBS, Biological Industries, Israel) as described previously [41]. In two-dimensional (2D) and three-dimensional (3D) models, the cells were grown on flat-bottomed and PrimeSurface 96U plates (Sumitomo Bakelite, Tokyo, Japan), respectively. CLDN2 promoter construct and internal control pRL-TK vector were transfected into the cells using HilyMax (Dojindo Laboratories, Kumamoto, Japan). The pTRE2 (mock) and CLDN2/pTRE2 mammalian expression vectors were transfected into spheroid cells using ScreenFect A (Fujifilm Wako Pure Chemical Industries).

### 4.3. Cell Viability Assay

The viability of 2D cultured cells was determined by a Cell Counting Kit-8 (CCK-8) (Dojindo Laboratories, Kumamoto, Japan). The absorbance of CCK-8 was measured using an iMark microplate absorbance reader (Bio-Rad Laboratories, Hercules, CA, USA). The viability of 3D cultured cells was determined by a CellTiter-Glo 3D Cell Viability Assay kit (Promega, Madison, WI, USA). The number of viable cells in 3D culture was quantified by ATP content. Viabilities of 2D and 3D cultured cells were calculated as percentage of the vehicle groups. The intensity of chemiluminescence was measured using a Luminescencer Octa AB-2270 (Atto, Tokyo, Japan). The cell viability was calculated as a percentage of the vehicle groups.

### 4.4. Sodium Dodecyl Sulfate-Polyacrylamide Gel Electrophoresis and Western Blotting

A549 cells (1 × 10^5^ /dish) were cultured on 6 cm dishes for 3 days. After treatment with chemicals, the cells were collected into cold phosphate-buffered saline, followed by solubilization with a lysis buffer. Sodium dodecyl sulfate-polyacrylamide gel electrophoresis and Western blotting were performed as described previously [41]. The optical band density was quantified by ImageJ software (National Institute of Health, Bethesda, MD, USA).

### 4.5. Cell Distribution of CLDN

A549 cells (5 × 10^4^ /dish) were cultured on 35 mm dishes laid with cover slips for 3 days. The fixation, permeabilization, blocking, and staining of antibodies were performed as described previously [41]. The cell distribution of CLDN1, CLDN2, and ZO-1 was analyzed using an LSM700 confocal laser microscope (Carl Zeiss, Jena, Germany).

### 4.6. Reverse Transcription and Quantitative Real-Time PCR

A549 cells (1 × 10^5^ /dish) were cultured on 6 cm dishes for 3 days. After treated with chemicals, the cells were lysed in TRI reagent (Molecular Research Center, Cincinnati, OH, USA). After isolation of total RNA, reverse transcription was performed using ReverTraAce qPCR RT Kit (Toyobo, Osaka, Japan). Quantitative real-time PCR was conducted with Eco Real-Time PCR system (AS One, Osaka, Japan) using Thunderbird SYBR qPCR Mix (Toyobo). Primer pairs are shown in Table 1. The threshold cycle (Ct) for each PCR product was calculated with the instrument’s software. The mRNA levels are represented as percentage of vehicle as described previously [22]

### 4.7. Spheroid Analysis

In 3D model, A549 cells (5 × 10^3^ /well) were cultured on PrimeSurface 96U plates using DMEM supplemented with 5% FBS for 3 days. The circumference length of spheroids was calculated using ImageJ software. The spheroids were incubated with LOX-1, a hypoxia probe, or DXR, an anthracycline anticancer drug for 1 h at 37 °C. The fluorescence intensities of LOX-1 and DXR were measured using a BZ-X810 fluorescence microscope (Keyence, Osaka, Japan). Apoptotic and necrotic cells were stained using an Apoptotic and Necrotic Cell Detection Kit (Takara Bio, Shiga, Japan), and the number of apoptotic and necrotic cells was calculated using a fluorescence microscope.

### 4.8. Statistical Analysis

Data are presented as means ± S.E.M. Comparisons between two groups were made using Student’s *t* test. Differences between groups were analyzed by one-way analysis of variance, and corrections for multiple comparison were made using Tukey’s multiple comparison test. Statistical analyses were performed using KaleidaGraph version 4.5.1 software (Synergy Software, PA, USA). Significant differences were assumed at *p* < 0.05.

## 5. Conclusions

We found that fisetin can decrease CLDN2 expression in A549 cells without affecting CLDN1 expression. The phosphorylation level of Akt, one regulator of CLDN2 expression, was decreased by fisetin, suggesting that fisetin decreases CLDN2 expression mediated by Akt inhibition. In contrast, CLDN1 expression may not be regulated by PI3K/Akt signal. Fisetin decreased the protein level of Nrf2, and mRNA levels of ABCB1 and ABCC1 transporters, which are involved in the efflux of anticancer drugs, in A549 spheroids. The anticancer drug-induced toxicity in the spheroids was enhanced by fisetin. The fisetin-induced elevation of anticancer drug-induced toxicity is suggested by the reduction of CLDN2 expression mediated through the downregulation of Nrf2 stress signal and/or expression of ABCB1 and ABCC1 transporters. We need further studies to clarify the regulatory mechanisms of fisetin on anticancer drug-induced toxicity. Our data indicated that foods and supplements rich in fisetin may be useful to prevent the malignant progression of lung adenocarcinoma.

## Figures and Tables

**Figure 1 ijms-23-07536-f001:**
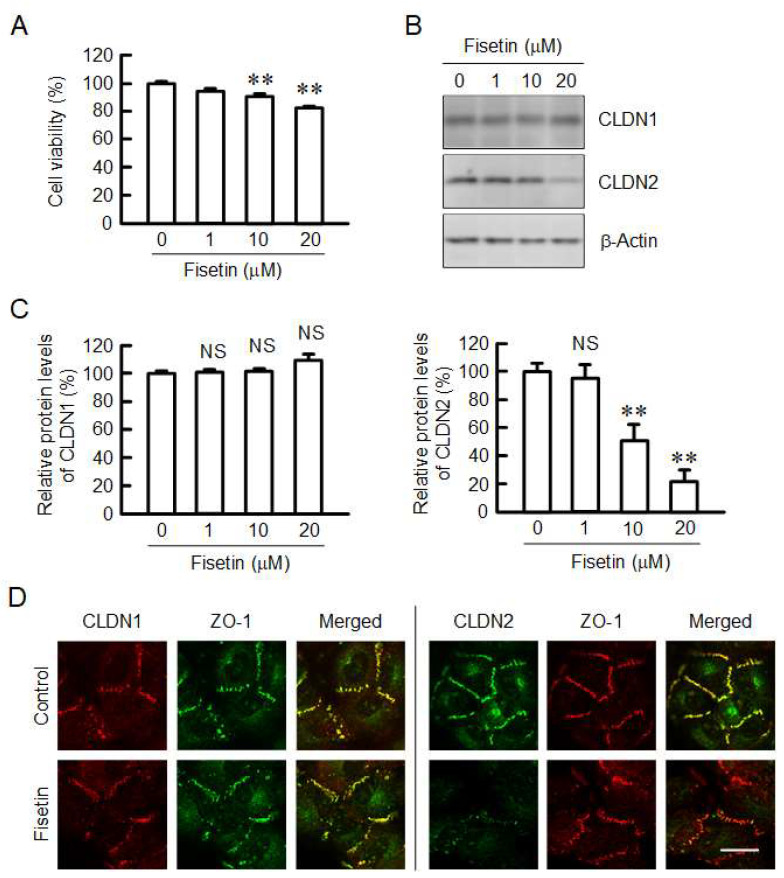
Decrease in protein level of CLDN2 by fisetin in A549 cells. (**A**) A549 cells were incubated with 0, 1, 10, and 20 μM fisetin for 24 h. Cell viability is represented as a percentage of 0 μM. (**B**,**C**) The cell lysates were immunoblotted with anti-CLDN1, anti-CLDN2, and anti-β-actin antibodies. After normalization by β-actin expression, the protein levels of CLDN1 and CLDN2 are represented as a percentage of 0 μM. (**D**) Cells cultured on cover glasses were incubated in the absence (control) and presence of 20 μM fisetin for 24 h. Control cells were treated with DMSO as a vehicle. The concentration of DMSO in the control and fisetin-treated cells was 0.1%. The cells were stained with rabbit anti-CLDN1 (red) plus mouse anti-ZO-1 (green), or mouse anti-CLDN2 (green) plus rabbit anti-ZO-1 (red) antibodies. Merged images are shown on the right. Scale bar indicates 10 μM. n = 3–6. ** *p* < 0.01 and ^NS^
*p* > 0.05 compared with 0 μM.

**Figure 2 ijms-23-07536-f002:**
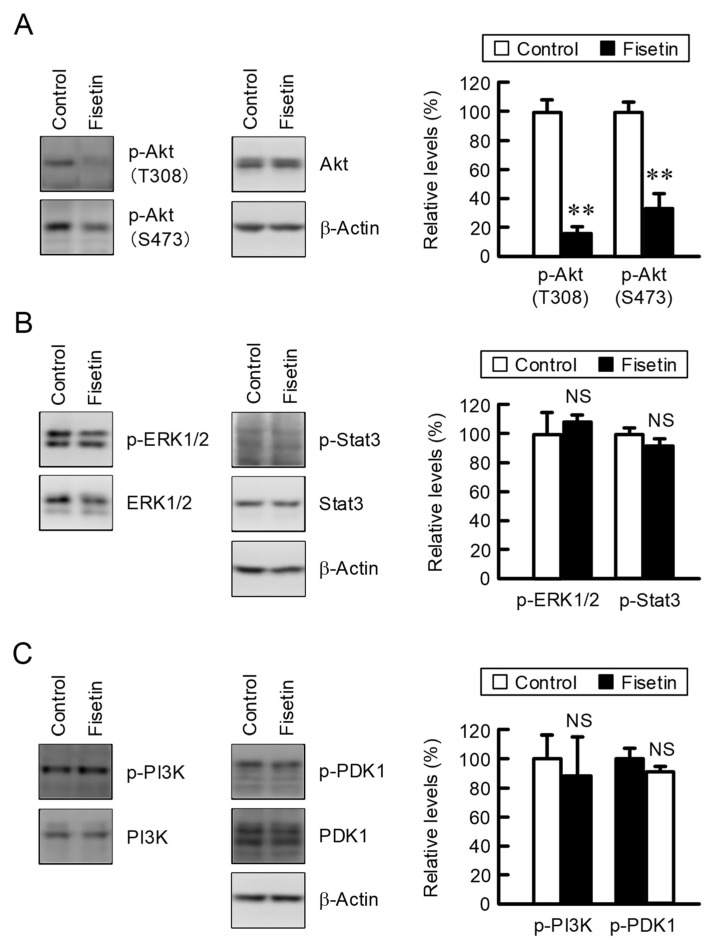
Effects of fisetin on the phosphorylation of intracellular signaling molecules. (**A**–**C**) Cells were incubated in the absence (control) and presence of 20 μM fisetin for 2 h. Control cells were treated with DMSO as a vehicle. The concentration of DMSO in the control and fisetin-treated cells was 0.1%. The protein levels of p-Akt (T308), p-Akt (S473), Akt, p-ERK1/2, ERK1/2, p-Stat3, Stat3, p-PI3K, PI3K, p-PDK1, PDK1, and β-actin were examined by Western blotting. The expression levels of p-Akt, p-ERK1/2, p-PI3K, and p-PDK1 were represented as a percentage of control. n = 3–4. ** *p* < 0.01 and ^NS^
*p* > 0.05 compared with control.

**Figure 3 ijms-23-07536-f003:**
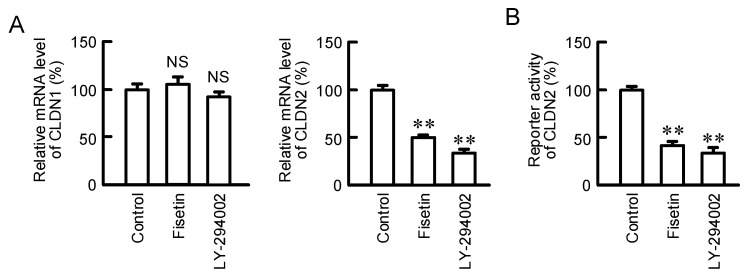
Decrease in mRNA level of CLDN2 by fisetin and LY-294002 in A549 cells. (**A**) A549 cells were incubated in the absence (control) and presence of 20 μM fisetin or 10 μM LY-294002 for 6 h. Control cells were treated with DMSO as a vehicle. The concentration of DMSO in the control and fisetin-treated cells was 0.1%. Real-time PCR was performed using primer pairs for CLDN1, CLDN2, and β-actin. After normalized by β-actin, the mRNA levels of CLDN1 and CLDN2 are represented as a percentage of control. (**B**) The cells were transfected with CLDN2 promoter construct and internal control pRL-TK vector. The reporter activity was measured using Dual-Glo luciferase assay kit and represented as a percentage of control. n = 3–4. ** *p* < 0.01 and ^NS^
*p* > 0.05 compared with control.

**Figure 4 ijms-23-07536-f004:**
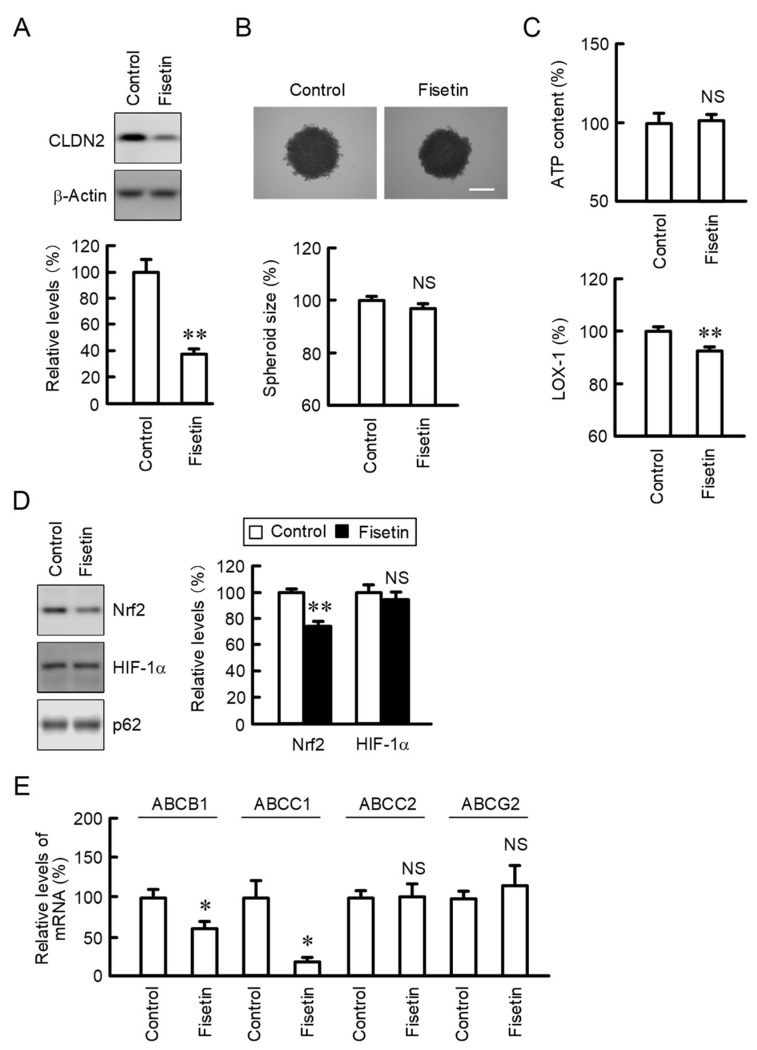
Effect of fisetin on property of spheroids. A549 cells were cultured in the round-bottom plates for 96 h and the cells formed spheroids. The spheroids were incubated in the absence (control) and presence of 20 μM fisetin for 24 h. Control cells were treated with DMSO as a vehicle. The concentration of DMSO in the control and fisetin-treated cells was 0.1%. (**A**) The protein levels of CLDN2 and β-actin were examined by Western blotting. The expression levels of CLDN2 were represented as a percentage of control. (**B**) Upper images are shown in upper panel. Scale bar indicates 500 μM. Spheroid size is represented as a percentage of control. (**C**) ATP content and fluorescence intensity of LOX-1 are represented as a percentage of control. (**D**) The protein levels of HIF-1α, Nrf2, and nucleoporin p62 (p62) were examined by Western blotting. The expression levels of HIF-1α and Nrf2 were represented as a percentage of control. (**E**) The mRNA levels of ABCB1, ABCC1, ABCC2, and ABCG2 were measured by real-time PCR and represented as a percentage of control. n = 5–6. ** *p* < 0.01, * *p* < 0.05, and ^NS^
*p* > 0.05 compared with control.

**Figure 5 ijms-23-07536-f005:**
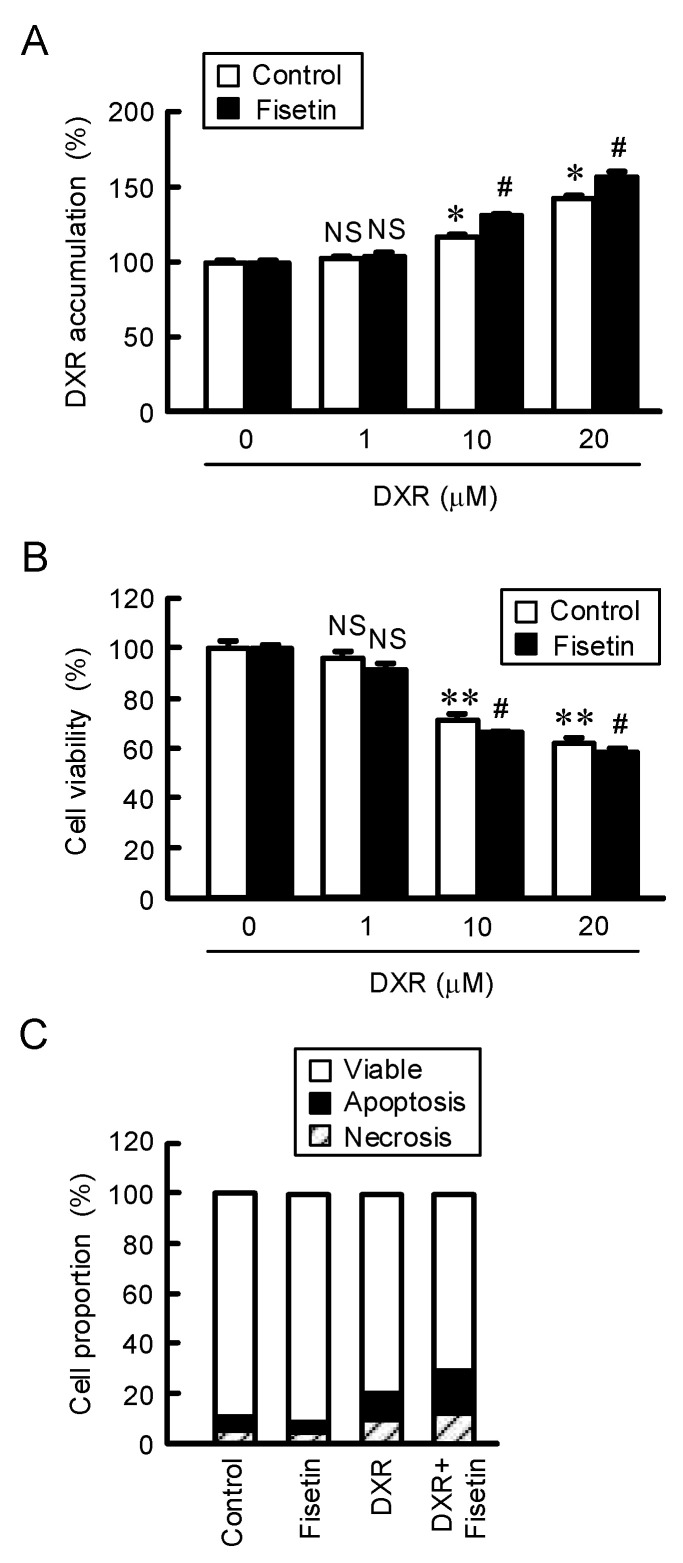
Increase in accumulation and toxicity of DXR by fisetin in spheroids. (**A**,**B**) The spheroids were treated with DXR in the absence (control) and presence of 20 μM fisetin for (**A**) 1 h or (**B**) 24 h. Control cells were treated with DMSO as a vehicle. The concentration of DMSO in the control and fisetin-treated cells was 0.1%. The fluorescence intensity of DXR and cell viability are represented as a percentage of 0 μM DXR. (**C**) The spheroids were treated in the absence (control) and presence of 20 μM fisetin, 20 μM DXR, or both of them for 24 h. The proportions of viable cells and apoptotic and necrotic cell deaths are represented as a percentage of total cells. n = 4–6. ** *p* < 0.01 and * *p* < 0.05 compared with 0 μM DXR. ^#^
*p* < 0.05 compared with DXR alone. ^NS^
*p* > 0.05 compared with 0 μM DXR or control.

**Figure 6 ijms-23-07536-f006:**
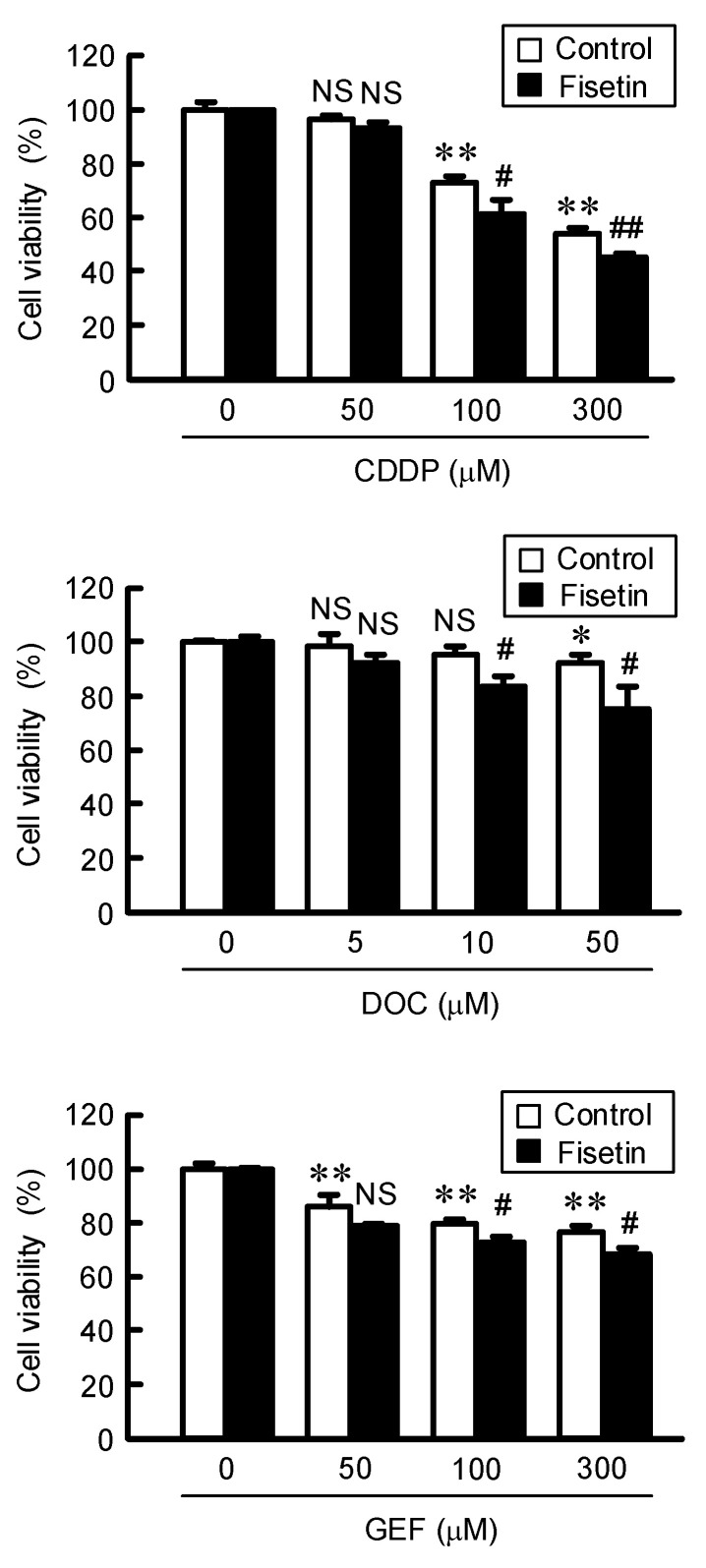
Enhancement of anticancer drug-induced toxicity by fisetin in spheroids. In the absence (control) and presence of 20 μM fisetin, the spheroids were treated with CDDP, DOC, and GEF at indicated concentration for 24 h. Control cells were treated with DMSO as a vehicle. The concentration of DMSO in the control and fisetin-treated cells was 0.1%. The cell viability was measured using a CellTiter-Glo 3D Cell Viability Assay kit and represented as a percentage of 0 μM anticancer drugs. n = 5–6. ** *p* < 0.01 and * *p* < 0.05. ^##^
*p* < 0.01 and ^#^
*p* < 0.05 compared with control. ^NS^
*p* > 0.05 compared with 0 μM DXR or control.

**Figure 7 ijms-23-07536-f007:**
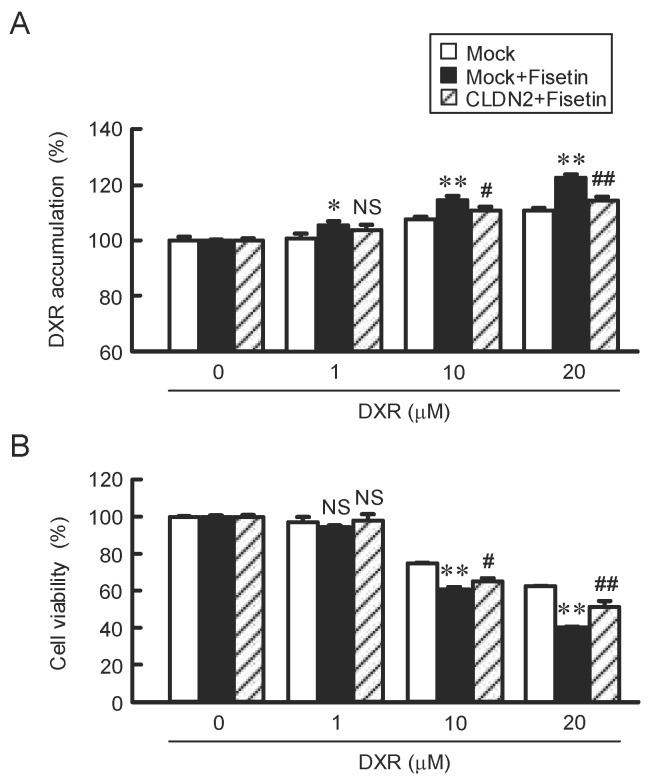
Inhibition of elevation of fisetin-induced anticancer drug toxicity by CLDN2 overexpression. The spheroid cells transfected with mock or CLDN2/pTRE2 expression vector were incubated in the absence (control) and presence of 20 μM fisetin for 24 h. Control cells were treated with DMSO as a vehicle. The concentration of DMSO in the control and fisetin-treated cells was 0.1%. (**A**) The cells were treated with DXR for 1 h. The fluorescence intensity of DXR in the spheroids is represented as a percentage of 0 μM DXR. (**B**) The cells were treated with DXR at indicated concentration for 24 h. Cell viability is represented as a percentage of 0 μM DXR. n = 5–6. ** *p* < 0.01 and * *p* < 0.05 compared with 0 μM DXR or mock + fisetin. ^##^
*p* < 0.01 and ^#^
*p* < 0.05 compared with mock + fisetin. ^NS^
*p* > 0.05 compared with mock or mock + fisetin.

**Figure 8 ijms-23-07536-f008:**
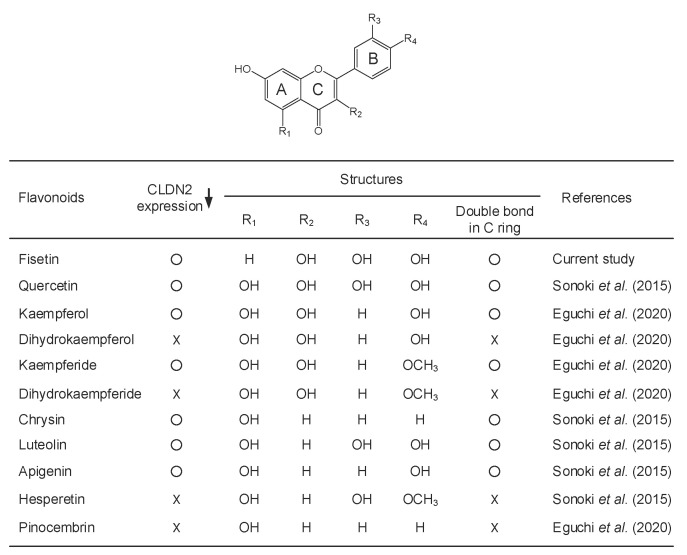
The relationship between structural characters and function of flavonoids. Representative chemical structure of flavonoids is shown above. The substituents at each position of R1, R2, R3, and R4 on the A-, B-, or C-ring in flavonoids are presented as hydrogen (H), hydroxy (OH), and methoxy (OCH_3_). In the CLDN2 expression column, the flavonoids that have the ability to reduce the expression of CLDN2 are represented as “○” and those do not have the ability are represented as “x”. In the double bound in C ring column, the flavonoids that have double bond are represented as “○” and those do not have the double bond are represented as “x” [23,24].

**Table 1 ijms-23-07536-t001:** Primer pairs for real-time PCR.

Genes	Direction	Sequence (5′→3′)
*CLDN1*	Sense	ATGAGGATGGCTGTCATTGG
	Antisense	ATTGACTGGGGTCATAGGGT
*CLDN2*	Sense	ATTGTGACAGCAGTTGGCTT
	Antisense	CTATAGATGTCACACTGGGTGATG
*HIF-1a*	Sense	AACGTCGAAAAGAAAAGTCTCG
	Antisense	AAATCACCAGCATCCAGAAGTT
*Nrf2*	Sense	TCCAGTCAGAAACCAGTGGAT
	Antisense	GAATGTCTGCGCCAAAAGCTG
*ABCB1*	Sense	CCCATCATTGCAATAGCAGG
	Antisense	TGTTCAAACTTCTGCTCCTGA
*ABCC1*	Sense	ATGTCACGTGGAATACCAGC
	Antisense	GAAGACTGAACTCCCTTCCT
*ABCC2*	Sense	ACAGAGGCTGGTGGCAACC
	Antisense	ACCATTACCTTGTCACTGTCCATGA
*ABCG2*	Sense	AGATGGGTTTCCAAGCGTTCAT
	Antisense	CCAGTCCCAGTACGACTGTGACA
*β-Actin*	Sense	CCTGAGGCACTCTTCCAGCCTT
	Antisense	TGCGGATGTCCACGTCACACTTC

## Data Availability

Not applicable.

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
