# Peer review of "Increase in Anticancer Drug-Induced Toxicity by Fisetin in Lung Adenocarcinoma A549 Spheroid Cells Mediated by the Reduction of Claudin-2 Expression"

_ijms, 2022, doi:10.3390/ijms23147536_

Round 1
Reviewer 1 Report
In this study, the authors have shown that fisetin, a dietary flavonoid, can reduce the expression of claudin 2 in A549 lung adenocarcinoma cells through downregulation of Akt phosphorylation. Using a 3D spheroid model of A549, fisetin is also shown to enhance the cytotoxic effects of anticancer agents such as doxorubicin, cisplatin, docetaxel, and gefitinib, while claudin 2 overexpression reduces this effect. Following are some comments that can help improve the quality of the work presented:
1. Since the title of the manuscript mentions that fisetin attenuates chemoresistance of lung adenocarcinoma spheroids mediated by reduction of claudin 2 expression, it will be meaningful to show the expression levels of claudin 2 in A549 spheroids subjected to treatment with fisetin.
2. Are the A549 cells used in this study resistant to doxorubicin or other anticancer drugs tested? If this spheroid model is not chemoresistant then at best, it may be concluded that cotreatment with fisetin enhances the sensitivity of A549 spheroids towards doxorubicin, etc.
3. In figure 2, loading controls should be included in the Western blots.
4. In figure 3, LY-294002, a PI3K inhibitor is used. But the abstract (line 20) refers to Akt inhibitor.
5. In the abstract (lines 21-22), it is mentioned that fisetin significantly decreased hypoxia level and hypoxia inducible factor-1alpha expression in spheroids. However, the results in Figure 4 clearly demonstrate that the protein levels of HIF-1a do not decrease due to treatment with fisetin. The reduction in LOX-1, although shown to be statistically significant, does not appear considerable (~ 5%) to conclude that fisetin reduces hypoxia levels.
6. cis-diamminedichloride platinum (CDDP) can be referred to as ‘cisplatin’, especially since all other anticancer drugs have been mentioned with their generic names.
7. The concentration of fisetin is missing in the legend to figure 6.
8. Also, in the legend to figure 6, 0 uM DXR is mentioned. However, the experiment shown in the figure does not have DXR.
9. Please replace ‘as indicated concentration’ with ‘at indicated concentration’ in figure 7 legend.
10. In the Discussion’s first sentence (line 211-212), CLDN2 is reported to be highly expressed in human lung adenocarcinoma. However, one of the cell lines mentioned is a prostate cancer cell line (PC-3).
11. Line 260-261 - sensitivity of A549 spheroid cells against CCDP, etc was enhanced by DXR?
12. Please mention the culture media used to generate 3D spheroids from the A549 cell line in the methods.
Author Response
We thank you very much for your careful reading of our manuscript and valuable comments.
Comment 1
Since the title of the manuscript mentions that fisetin attenuates chemoresistance of lung adenocarcinoma spheroids mediated by reduction of claudin 2 expression, it will be meaningful to show the expression levels of claudin 2 in A549 spheroids subjected to treatment with fisetin.
Answer
Following your suggestion, we performed additional experiments. Claudin-2 expression level was decreased by fisetin in A549 spheroids. Please see new figure 4A.
Comment 2
Are the A549 cells used in this study resistant to doxorubicin or other anticancer drugs tested? If this spheroid model is not chemoresistant then at best, it may be concluded that cotreatment with fisetin enhances the sensitivity of A549 spheroids towards doxorubicin, etc.
Answer
The A549 cells used in this study is not resistant to anticancer drugs. Following your suggestion, we modified the description from “chemoresistance” to “chemosensitivity”.
Comment 3
In figure 2, loading controls should be included in the Western blots.
Answer
Following your suggestion, we showed the data of b-actin as the loading control. Please see new figure 2.
Comment 4
In figure 3, LY-294002, a PI3K inhibitor is used. But the abstract (line 20) refers to Akt inhibitor.
Answer
Thank you very much for your indication. We corrected it.
Comment 5
In the abstract (lines 21-22), it is mentioned that fisetin significantly decreased hypoxia level and hypoxia inducible factor-1alpha expression in spheroids. However, the results in Figure 4 clearly demonstrate that the protein levels of HIF-1a do not decrease due to treatment with fisetin. The reduction in LOX-1, although shown to be statistically significant, does not appear considerable (~ 5%) to conclude that fisetin reduces hypoxia levels.
Answer
Thank you very much for your indication. We corrected it.
Comment 6
Cis-diamminedichloride platinum (CDDP) can be referred to as ‘cisplatin’, especially since all other anticancer drugs have been mentioned with their generic names.
Answer
Following your suggestion, we modified it.
Comment 7
The concentration of fisetin is missing in the legend to figure 6.
Answer
Thank you very much for your indication. We added the concentration of fisetin.
Comment 8
Also, in the legend to figure 6, 0 uM DXR is mentioned. However, the experiment shown in the figure does not have DXR.
Answer
Thank you very much for your indication. We corrected it.
Comment 9
Please replace ‘as indicated concentration’ with ‘at indicated concentration’ in figure 7 legend.
Answer
Following your suggestion. We modified it.
Comment 10
In the Discussion’s first sentence (line 211-212), CLDN2 is reported to be highly expressed in human lung adenocarcinoma. However, one of the cell lines mentioned is a prostate cancer cell line (PC-3).
Answer
PC-3 cells derived from prostate cancer are very famous. But we used human lung adenocarcinoma-derived PC-3 cells in the present study. Please see below information. We added the catalog number of PC-3 cells.
https://cellbank.nibiohn.go.jp/~cellbank/cgi-bin/search_res_det.cgi?ID=252
Comment 11
Line 260-261 - sensitivity of A549 spheroid cells against CCDP, etc was enhanced by DXR?
Answer
Thank you very much for your indication. The sensitivity of A549 spheroid cells against anticancer drugs was not enhanced by DXR. We corrected it.
Comment 12
Please mention the culture media used to generate 3D spheroids from the A549 cell line in the methods.
Answer
Following your suggestion. We described the culture media used in 3D spheroids. Please see line 335.
Reviewer 2 Report
The manuscript ID entitled "Fisetin attenuates chemoresistance of lung adenocarcinoma 2 A549 spheroid cells mediated by reduction of claudin-2 expression" is a good study. Claudins (CLDNs) are components of tight junction in epithelial cells and regulate para-13 cellular fluxes of mineral ions and low molecular compounds. CLDN2 is absent in normal lung tissues but is highly expressed in adenocarcinoma tissues. The reduction of CLDN2 expression res-15 cues chemoresistance in a spheroid model of A549 cells derived from human lung adenocarcinoma. 16 Here, we found that CLDN2 expression is decreased by fisetin, a dietary flavonoid, in A549 cells. Therefore, the expression mechanism and chemosensitivity were investigated by real-time polymerase chain reaction, Western blotting, and viability assay analyses. Fisetin decreased phosphorylated 19 Akt level and CLDN2 expression was decreased by an Akt inhibitor, suggesting that Akt is involved 20 in the reduction of CLDN2 expression by fisetin. Fisetin significantly decreased hypoxia level and hypoxia-inducible factor-1 expression in the spheroids. The proportion of both apoptotic and necrotic dead cells was increased by doxorubicin, an anthracycline antibiotic, which was enhanced by fisetin. Similarly, the cytotoxic effects of other anticancer drugs including cis-diamminedichloride platinum, docetaxel, and gefitinib were enhanced by fisetin. The fisetin-induced rescue of chemoresistance was inhibited by the overexpression of CLDN2. These results suggest that fisetin can attenuate chemoresistance in A549 spheroid cells mediated by the reduction of CLDN2 expression. However, the following points are to be addressed,
The results of the abstract are unclear. The author may be mentioned that fold change for expression level and IC50 value for inhibition make more sense.
The correlation lacking in the background section between chemoresistance and hypoxia in the tumor environment
Reverse transcription and quantitative real-time PCR------reaction conditions are missing
Author Response
We thank you very much for your careful reading of our manuscript and valuable comments.
Comment 1
The results of the abstract are unclear. The author may be mentioned that fold change for expression level and IC50 value for inhibition make more sense.
Answer
Following your suggestion, we modified the results of Abstract section.
Comment 2
The correlation lacking in the background section between chemoresistance and hypoxia in the tumor environment.
Answer
We described the correlation between chemoresistance and hypoxia in the tumor environment in the Introduction. Please see line 39.
Comment 3
Reverse transcription and quantitative real-time PCR------reaction conditions are missing
Answer
Following your suggestion, we described the reaction conditions. Please see line 324.
Round 2
Reviewer 1 Report
Line 189, remove DXR.
It is surprising to note that A549 spheroids were cultured using the same media as used for 2D culture (DMEM supplemented with 5% FBS). Most studies use special tumorsphere forming media or regular media supplemented with different growth factors (EGF, FGF, etc).
Adding some images of the spheroids if available will strengthen the existing data.
Author Response
We thank you very much for your careful reading of our manuscript and valuable comments.
Comment 1
It is surprising to note that A549 spheroids were cultured using the same media as used for 2D culture (DMEM supplemented with 5% FBS). Most studies use special tumorsphere forming media or regular media supplemented with different growth factors (EGF, FGF, etc).
Adding some images of the spheroids if available will strengthen the existing data.
Answer
Following your suggestion, we showed the representative images of A549 spheroids. Please see new figure 4. Some investigators reported that A549 cells can form spheroids using normal medium in PrimeSurface plates (Sumitomo Bakelite) without supplementation with different growth factors.
Muguruma M. et al., Biochem. Biophys. Res. Commun., 533, 268-274 (2020)
Kulkarni A. et al., Nature Communications, 12, 3834 (2021)